# GraphSAD: Learning Graph Representations with Structure-Attribute Disentanglement

## Abstract

Graph Neural Networks (GNNs) learn effective node/graph representations by aggregating the attributes of neighboring nodes, which commonly derives a single representation mixing the information of *graph structure* and *node attributes*. However, these two kinds of information might be semantically inconsistent and could be useful for different tasks. In this paper, we aim at learning node/graph representations with **S**tructure-**A**ttribute **D**isentanglement (*GraphSAD*). We propose to disentangle graph structure and node attributes into two distinct sets of representations, and such disentanglement can be done in either the input or the embedding space. We further design a metric to quantify the extent of such a disentanglement. Extensive experiments on multiple datasets show that our approach can indeed disentangle the semantics of graph structure and node attributes, and it achieves superior performance on both node and graph classification tasks.

## 1 Introduction

Representing nodes or entire graphs with informative low-dimensional feature vectors plays a crucial role in many real-world applications and domains, *e.g.* user analysis in social networks (Tan et al., 2011; Yan et al., 2013), relational inference in knowledge graphs (Bordes et al., 2013; Trouillon et al., 2016; Sun et al., 2019), molecular property prediction in drug/material discovery (Gilmer et al., 2017; Wu et al., 2018) and circuit response prediction in circuit design (Zhang et al., 2019). Recently, Graph Neural Networks (GNNs) (Kipf & Welling, 2017; Velickovic et al., 2018; Xu et al., 2019) have shown their superiority in many different tasks. In general, the essential idea of these methods is to learn effective node representations (or graph representations with an additional graph pooling) through aggregating the attributes of each node and its neighbors in an iterative and nonlinear way.

For an attributed graph, GNNs commonly encode the information of its *graph structure* and *node attributes* into a single representation. This might be problematic, since the semantic space of graph structure and node attributes might not be well aligned, and these two types of information could be useful for different tasks. For example, predicting the health condition of a user mainly depends on his/her profile information, and the social network does not provide too much meaningful information; in another case, the prediction of a user's social class mainly relies on his/her social network structure. Therefore, a more reasonable solution is to disentangle these two types of information into two distinct sets of representations, and the importance of which can be further determined by downstream tasks. Such disentangled representation has been proved to be beneficial to model's generalization ability and interpretability (Chen et al., 2016; Higgins et al., 2017; Alemi et al., 2017).

Recently, DisenGNN (Ma et al., 2019) studied disentangled node representation learning by grouping the neighbors of each node to different channels, and each channel corresponds to a different latent factor. In other words, DisenGNN focuses on disentangling the various latent factors of graph structure. By contrast, our work intends to disentangle the representations of graph structure and node attributes, which is orthogonal to their work and also more general.

In this paper, we aim to learn node/graph representations with **S**tructure-**A**ttribute **D**isentanglement (*GraphSAD*). As a naive trial, we first attempt to conduct disentanglement in the input space, named as *Input-SAD*, which separates a graph into a structure and an attribute component and then encodes these two components respectively. However, since graph structure and node attributes are not completely independent, it is better to suppress the dependency of these two factors in the embedding space, instead of directly separating the input graph. Inspired by this fact, we propose to distill a

graph's structure and attribute information into the distinct channels of embedding vectors, named as *Embed-SAD*. Concretely, for each node embedding, half of its elements capture the graph structure through edge reconstruction, and the other half extracts the attribute information by minimizing the mutual information with the structure counterpart and, at the same time, preserving semantic discriminability. In addition, we devise a metric to quantitatively evaluate graph representation's structure-attribute disentanglement, denoted as *SAD-Metric*, which measures the sensitivity of a model when varying either the graph structure or node attributes of an input graph.

We summarize our contributions as follows:

- We study structure-attribute disentangled node/graph representation learning through separating graph structure and node attributes in either the input or the embedding space.

- We design a quantitative metric to measure the extent of structure-attribute disentanglement, which is novel on its graph-specific data processing scheme.

- Through combining the proposed disentangling techniques with various GNNs, we empirically verify our method's superior performance on both the node and graph classification benchmark datasets. Also, we analyze the disentangled graph representations via the proposed metric and qualitative visualization.

## 2 PROBLEM DEFINITION AND PRELIMINARIES

### 2.1 PROBLEM DEFINITION

We study learning node representations (*e.g.* social networks) or whole-graph representations (*e.g.* molecular graphs) of attributed graphs. Formally, we denote an attributed graph as $\mathcal{G} = (\mathcal{V}, \mathcal{E}, \mathcal{A})$. $\mathcal{V}$ denotes the set of nodes. $\mathcal{E} = \{(u, v, t_{uv})\}$ is the set of edges with $t_{uv}$ as the type of the edge connecting node $u$ and $v$ (*e.g.* different types of bonds in molecular graphs). $\mathcal{A} = \{\mathcal{A}_v | v \in \mathcal{V}\}$ represents the set of node attributes.

Our goal is to learn meaningful representations for each node or the whole graph. Existing GNNs typically mix both the graph structure and node attributes into a unified representation through neural message passing. However, in practice, these two types of information may encode different semantics and be useful for different tasks. Take the prediction on social networks as an example. When predicting the social class of users, the graph structure plays a more important role than user attributes, while user attributes are definitely more informative than graph structure when forecasting users' health conditions. It is therefore desirable to disentangle the information of graph structure and node attributes into different sets of representations and use the downstream task to determine their importance. Specifically, we define our problem as follows:

**Node/Graph Representation Learning with Structure-Attribute Disentanglement.** Given an attributed graph $\mathcal{G} = (\mathcal{V}, \mathcal{E}, \mathcal{A})$, we aim to learn node (or whole-graph) representations by disentangling the semantics of graph structure $\mathcal{S} = \{\mathcal{V}, \mathcal{E}\}$ and node attributes $\mathcal{A}$ into two distinct sets of representations, *i.e.* $z_v = [z_{v,\mathcal{S}}, z_{v,\mathcal{A}}]$ (or $z_{\mathcal{G}} = [z_{\mathcal{G},\mathcal{S}}, z_{\mathcal{G},\mathcal{A}}]$). The importance of the two kinds of representations is further determined by the downstream task such as node or graph classification.

### 2.2 PRELIMINARIES

**Graph Neural Networks (GNNs).** A GNN maps each node $v \in \mathcal{V}$ to an embedding vector $z_v$ and also encodes the entire graph $\mathcal{G}$ as vector $z_{\mathcal{G}}$. For an $L$-layer GNN, the $L$-hop information surrounding each node is captured via a neighborhood aggregation mechanism. Formally, the $l$-th GNN layer can be defined as:

$$z_v^{(l)} = \text{COMBINE}^{(l)}\left(z_v^{(l-1)}, \text{AGGREGATE}^{(l)}\left(\left\{\left(z_v^{(l-1)}, z_u^{(l-1)}, t_{uv}\right) : u \in \mathcal{N}(v)\right\}\right)\right), \quad (1)$$

where $\mathcal{N}(v)$ is the set of node $v$'s neighbors, $t_{uv}$ denotes edge attribute, $z_v^{(l)}$ denotes the representation of $v$ at the $l$-th layer, and $z_v^{(0)}$ is initialized by the node attribute $\mathcal{A}_v$. Using all the node embeddings in a graph, the entire graph's embedding can be derived by a permutation-invariant readout function:

$$z_{\mathcal{G}} = \text{READOUT}\left(\{z_v | v \in \mathcal{V}\}\right). \quad (2)$$

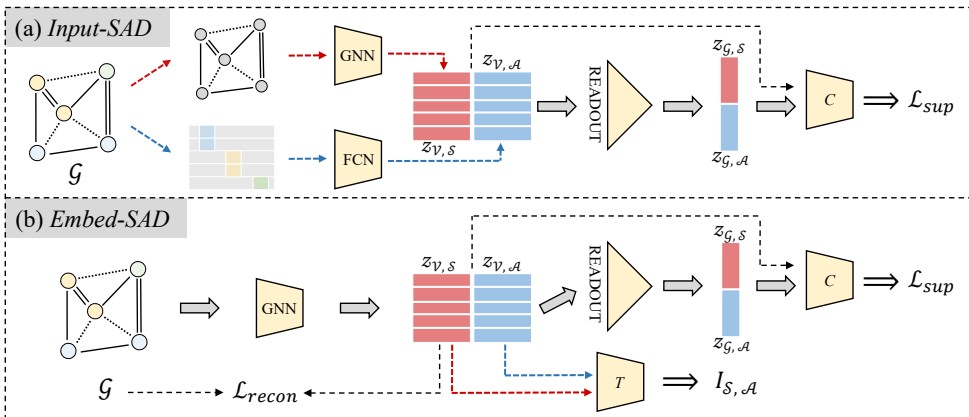

Figure 1: (a) *Input-SAD* model disentangles a graph into a structure and an attribute component and encodes them separately. (b) *Embed-SAD* model distills the structure and attribute information of a graph into two distinct channels of embedding vectors.

**Mutual Information Estimator.** Mutual information (MI) quantifies the mutual dependency between two random variables. Some recent works (Belghazi et al., 2018; Hjelm et al., 2019) studied neural-network-based MI estimators. Among which, the Noise-Contrastive Estimation (NCE) (Gutmann & Hyvärinen, 2010; 2012) was first employed as a lower bound of MI by van den Oord et al. (2018), and we also adopt this estimator in our method for its effectiveness and concision. In practice, for two random variables $\mathbf{x}_1$ and $\mathbf{x}_2$, given one positive pair $(\mathbf{x}_1^+, \mathbf{x}_2^+) \sim p(\mathbf{x}_1, \mathbf{x}_2)$ and $K$ distractors $(\mathbf{x}_1^+, \mathbf{x}_{2,j}) \sim p(\mathbf{x}_1)p(\mathbf{x}_2)$ $(j = 1, 2, \cdots, K)$, the NCE estimation of MI is defined as:

$$\mathcal{I}_{\text{NCE}}\big(\mathbf{x}_1^+, \mathbf{x}_2^+, \{\mathbf{x}_{2,j}\}_{j=1}^K\big) = \log(K+1) + \log \frac{\exp\big(T(\mathbf{x}_1^+, \mathbf{x}_2^+)\big)}{\exp\big(T(\mathbf{x}_1^+, \mathbf{x}_2^+)\big) + \sum_{j=1}^K \exp\big(T(\mathbf{x}_1^+, \mathbf{x}_{2,j})\big)}, \quad (3)$$

where $T(\cdot, \cdot)$ is a parameterized discriminator function which outputs a scalar value for a pair of input samples, and its architecture is detailed in Sec. 5.1.

## 3 LEARNING GRAPH REPRESENTATIONS WITH STRUCTURE-ATTRIBUTE DISENTANGLEMENT

### 3.1 INPUT-SAD: STRUCTURE-ATTRIBUTE DISENTANGLEMENT FOR INPUTS

As an initial attempt, we seek to learn structure-attribute disentangled node/graph representations by separating a graph into a structure and an attribute component and then encoding them respectively, as shown in Fig. 1(a). Concretely, given an attributed graph $\mathcal{G} = (\mathcal{V}, \mathcal{E}, \mathcal{A})$, these two components are constructed and encoded as follows.

The *structure component* extracts the graph structure and forms another graph $\mathcal{G}_{\mathcal{S}} = (\mathcal{V}_{\mathcal{S}}, \mathcal{E}_{\mathcal{S}}, \mathcal{A}_{\mathcal{S}})$, in which the node and edge sets remain unchanged, *i.e.* $\mathcal{V}_{\mathcal{S}} = \mathcal{V}$, $\mathcal{E}_{\mathcal{S}} = \mathcal{E}$, and the out-degree of each node serves as its attribute, *i.e.* $\mathcal{A}_{\mathcal{S}} = \{d(v) | v \in \mathcal{V}_{\mathcal{S}}\}$ ($d(\cdot)$ denotes the out-degree function). A GNN maps this component to a $\delta$-dimensional embedding space:

$$(z_{\mathcal{V},\mathcal{S}}, z_{\mathcal{G},\mathcal{S}}) = \text{GNN}(\mathcal{V}_{\mathcal{S}}, \mathcal{E}_{\mathcal{S}}, \mathcal{A}_{\mathcal{S}}), \quad (4)$$

where $z_{\mathcal{V},\mathcal{S}} = \{z_{v,\mathcal{S}} | v \in \mathcal{V}\} \in \mathbb{R}^{|\mathcal{V}| \times \delta}$ denotes the node embeddings derived only by the graph structure, and $z_{\mathcal{G},\mathcal{S}} \in \mathbb{R}^{\delta}$ is the embedding of the entire structure component.

The *attribute component* is formed as a feature matrix $\mathbf{U} \in \mathbb{R}^{|\mathcal{V}| \times D}$, where the feature vector $\mathbf{U}_v \in \mathbb{R}^D$ is a $D$-dimensional embedding of node attribute $\mathcal{A}_v$. For this component, a fully-connected network and a readout function (*e.g.* mean pooling in our implementation) are used for encoding:

$$z_{\mathcal{V},\mathcal{A}} = \text{FCN}(\mathbf{U}), \quad z_{\mathcal{G},\mathcal{A}} = \text{READOUT}(z_{\mathcal{V},\mathcal{A}}), \quad (5)$$

where $z_{\mathcal{V},\mathcal{A}} = \{z_{v,\mathcal{A}} | v \in \mathcal{V}\} \in \mathbb{R}^{|\mathcal{V}| \times \delta}$ denotes the attribute embeddings of the nodes in graph $\mathcal{G}$, and $z_{\mathcal{G},\mathcal{A}} \in \mathbb{R}^{\delta}$ embeds the whole attribute component.

The complete information of graph $\mathcal{G}$ is restored by concatenating the structure and attribute embedding for each node and for entire graph:

$$z_{\mathcal{V}} = \left\{ [z_{v,\mathcal{S}}, z_{v,\mathcal{A}}] \,|\, v \in \mathcal{V} \right\} \in \mathbb{R}^{|\mathcal{V}| \times 2\delta}, \quad z_{\mathcal{G}} = [z_{\mathcal{G},\mathcal{S}}, z_{\mathcal{G},\mathcal{A}}] \in \mathbb{R}^{2\delta}, \tag{6}$$

where $[\cdot, \cdot]$ denotes the concatenation operation. Upon these concatenated node/graph embeddings, the prediction task (*e.g.* node/graph classification) is performed by a task-specific network $C$, which defines the supervised loss $\mathcal{L}_{\text{sup}}$ for model optimization:

$$\min_{\text{GNN},\text{FCN},C} \mathcal{L}_{\text{sup}}. \tag{7}$$

## 3.2 Embed-SAD: Structure-Attribute Disentanglement for Embeddings

The explicit separation of an input graph into structure and attribute components forces the independent encoding of graph structure and node attributes. However, these two factors are not completely independent. For example, in a social network, the social connections of a person can provide useful information about his/her character, and vice versa. Therefore, the representations derived by Input-SAD may not fully capture the structure and attribute information of a graph. To tackle this shortcoming, we seek to perform encoding on the original graph and distill its structure and attribute information into distinct channels of embedding vectors, as illustrated in Fig. 1(b).

For an attributed graph $\mathcal{G} = (\mathcal{V}, \mathcal{E}, \mathcal{A})$, a GNN is employed to map the graph to a $2\delta$-dimensional embedding space:

$$(z_{\mathcal{V}}, z_{\mathcal{G}}) = \text{GNN}(\mathcal{V}, \mathcal{E}, \mathcal{A}), \tag{8}$$

where $z_{\mathcal{V}} = \left\{ z_v | v \in \mathcal{V} \right\} \in \mathbb{R}^{|\mathcal{V}| \times 2\delta}$ denotes node embeddings, and $z_{\mathcal{G}} \in \mathbb{R}^{2\delta}$ is the embedding of entire graph. We further divide these embeddings into two channels:

$$z_{\mathcal{G}} = [z_{\mathcal{G},\mathcal{S}}, z_{\mathcal{G},\mathcal{A}}], \quad z_v = [z_{v,\mathcal{S}}, z_{v,\mathcal{A}}], \; \forall v \in \mathcal{V}, \tag{9}$$

where $z_{v,\mathcal{S}}, z_{v,\mathcal{A}} \, (z_{\mathcal{G},\mathcal{S}}, z_{\mathcal{G},\mathcal{A}}) \in \mathbb{R}^{\delta}$ are the structure and attribute embedding of node $v$ (the whole graph $\mathcal{G}$). In order to distill the structure and attribute information to the corresponding channel, we propose two learning schemes.

**Learning structure embedding by edge reconstruction.** The node embeddings fully capturing the graph structure are supposed to be capable of reconstructing the edges of this graph. Specifically, using the structure embeddings of a pair of nodes, we expect to predict the existence and type of the edge between them, which defines the reconstruction constraint for learning structure embeddings:

$$\mathcal{L}_{\text{recon}} = -\mathbb{E}_{u \sim P_{\mathcal{V}}, v \sim P_{\mathcal{V}}} \sum_{i=0}^{N_t} \mathbb{1}_{[t_{uv}=i]} \cdot \log p(y = i | z_{u,\mathcal{S}}, z_{v,\mathcal{S}}), \tag{10}$$

where $P_{\mathcal{V}}$ denotes the uniform distribution over $\mathcal{V}$, $\mathbb{1}_{[t_{uv}=i]}$ is the indicator function judging whether the edge $(u, v)$ belongs to type $i$ ($i = 0$ denotes there is no edge), $N_t$ is the number of different edge types, and $p(y | z_{u,\mathcal{S}}, z_{v,\mathcal{S}})$ is modeled by a neural network $F$ which is detailedly discussed in Sec. B.

This objective function relates to the ones proposed in VGAE (Kipf & Welling, 2016) and Graph-SAGE (Hamilton et al., 2017), while it additionally constrains the reconstruction of edge type, which enables the structure embeddings to adequately capture the information of graph structure.

**Learning attribute embedding by mutual information (MI) minimization.** Now that the structure embedding is obtained, we would like to derive the attribute embedding which extracts the information of node attributes and suppresses that of graph structure. To achieve this goal, we employ a neural-network-based MI estimator (*i.e.* the NCE estimator in Sec. 2.2) to estimate and, simultaneously, minimize the dependency between structure and attribute embedding.

In specific, we denote the structure and attribute latent factor as two random variables, $z_{\mathcal{S}}$ and $z_{\mathcal{A}}$, and regard the structure and attribute embedding of each node as the sample from the corresponding marginal distribution, *i.e.* $z_{v,\mathcal{S}} \sim p(z_{\mathcal{S}})$, $z_{v,\mathcal{A}} \sim p(z_{\mathcal{A}})$ ($v \in \mathcal{V}$). For computing the NCE estimation of MI, we define the structure and attribute embedding of the same node as a positive pair, *i.e.* $(z_{v,\mathcal{S}}, z_{v,\mathcal{A}}) \sim p(z_{\mathcal{S}}, z_{\mathcal{A}})$ ($v \in \mathcal{V}$), and the embedding pair constituted by two different nodes serves as a distractor, *i.e.* $(z_{v,\mathcal{S}}, z_{w,\mathcal{A}}) \sim p(z_{\mathcal{S}})p(z_{\mathcal{A}})$ ($v \neq w, v, w \in \mathcal{V}$). With these notions, the estimated MI between two latent factors is defined as:

$$\mathcal{I}_{\mathcal{S},\mathcal{A}} = \mathbb{E}_{v \sim P_{\mathcal{V}}, w_j \sim P_{\mathcal{V} \setminus \{v\}}} \mathcal{I}_{\text{NCE}}\left(z_{v,\mathcal{S}}, z_{v,\mathcal{A}}, \{z_{w_j,\mathcal{A}}\}_{j=1}^{K}\right), \tag{11}$$

---

**Algorithm 1:** Evaluation procedure for SAD-Metric.

---

**Input:** Evaluation set $\mathbb{D} = \{\mathcal{G}_i\}_{i=1}^N$, the model to be evaluated $\mathcal{M}$.
**Output:** The evaluation score for SAD-Metric.
Initialize the counter: $c = 0$ ;
**for** *i=1 to N* **do**
    $y_{\mathcal{G}_i} \leftarrow \text{RandomSample}(\{0, 1\})$ ;                      # Sample a binary label
    **if** $y_{\mathcal{G}_i} = 0$ **then**
        |  $\mathcal{G}_i' \leftarrow \text{ModifyAttribute}(\mathcal{G}_i)$ ;            # Modify the attribute factor of a graph
    **else**
        |__ $\mathcal{G}_i' \leftarrow \text{ModifyStructure}(\mathcal{G}_i)$ ;            # Modify the structure factor of a graph
    $z_{\mathcal{G}_i} \leftarrow \mathcal{M}(\mathcal{G}_i), z_{\mathcal{G}_i'} \leftarrow \mathcal{M}(\mathcal{G}_i')$ ;           # Infer graph embeddings
    $\Delta z_{\mathcal{G}_i} = |z_{\mathcal{G}_i} - z_{\mathcal{G}_i'}|$ ;                  # Compute embedding difference
    Predict the binary label $\hat{y}_{\mathcal{G}_i} = \arg\max_y p(y|\Delta z_{\mathcal{G}_i})$ ;
    **if** $\hat{y}_{\mathcal{G}_i} = y_{\mathcal{G}_i}$ **then**
        |__ $c \leftarrow c + 1$ ;

**Return** $\text{Acc} = c/N \times 100\%$ ;           # Compute prediction accuracy as SAD-Metric

---

where $P_{\mathcal{V}}$ and $P_{\mathcal{V}\setminus\{v\}}$ denote the uniform distribution over the node set with and without node $v$, and $\mathcal{I}_{\text{NCE}}(\cdot, \cdot, \cdot)$ is the NCE estimation function defined in Eq. 3.

Through minimizing this estimated MI, both the linear and nonlinear dependency between structure and attribute embeddings can be suppressed, which facilitates structure-attribute disentangled node/graph representations. Note that, such learning mechanism relates to the information bottleneck (IB) principle (Tishby et al., 2000; Tishby & Zaslavsky, 2015; Alemi et al., 2017), while, compared with IB, the proposed approach intends to separate two different sources of information instead of pursuing the maximal compression about the input.

**Model optimization.** We perform the prediction task (*e.g.* node/graph classification) by appending a task-specific network $C$ upon the disentangled node/graph embeddings (*i.e.* $z_{\mathcal{V}}$ or $z_{\mathcal{G}}$), which defines a supervised loss $\mathcal{L}_{\text{sup}}$ intended to be minimized. This supervised task also guarantees that the meaningful semantic information is not eliminated by the MI minimization scheme. For structure-attribute disentanglement, on one hand, the reconstruction loss $\mathcal{L}_{\text{recon}}$ is minimized to distill the information of graph structure into structure embeddings. On the other hand, the MI minimization is conducted in an adversarial manner, in which the discriminator $T$ (defined in Eq. 3) is trained to maximize $\mathcal{I}_{\mathcal{S},\mathcal{A}}$, while the GNN encoder seeks to minimize that term. The overall objective is:

$$\min_{\text{GNN},F,C} \max_T \mathcal{L}_{\text{sup}} + \lambda_1 \mathcal{L}_{\text{recon}} + \lambda_2 \mathcal{I}_{\mathcal{S},\mathcal{A}}, \tag{12}$$

where $\lambda_1$ and $\lambda_2$ are the trade-off parameters balancing between different objectives.

### 3.3 SAD-Metric: Structure-Attribute Disentanglement Metric

In order to quantify the extent of structure-attribute disentanglement achieved by various models, we devise a classifier-based metric to measure the learnt graph representations. Inspired by a previous work (Higgins et al., 2017), we focus on two desired properties of the disentangled representations: (1) *independence*: a representation vector is expected to be divided into several channels whose interdependency is as low as possible; (2) *interpretability*: each of these channels corresponds to a single latent factor of the data.

To derive such a metric, for a given graph $\mathcal{G}$, we first sample a binary label $y_{\mathcal{G}}$ from the uniform distribution over $\{0, 1\}$. According to this label, either the graph's structure or attribute is modified under the other factor fixed ($y_{\mathcal{G}} = 0$: fix $\mathcal{G}$'s structure; $y_{\mathcal{G}} = 1$: fix $\mathcal{G}$'s attribute), which forms the counterpart graph $\mathcal{G}'$. In practice, we modify a graph's structure through randomly dropping one of its edge (Rong et al., 2019), and the graph's attribute is modified by randomly altering the attribute of a node (implementation details are stated in Sec. B). Using the model to be evaluated, the embeddings of graph $\mathcal{G}$ and $\mathcal{G}'$ (*i.e.* $z_{\mathcal{G}}$ and $z_{\mathcal{G}'}$) are inferred, and we denote the absolute difference between these two embeddings as $\Delta z_{\mathcal{G}}$ (*i.e.* $\Delta z_{\mathcal{G}} = |z_{\mathcal{G}} - z_{\mathcal{G}'}|$). When the structure and attribute

Figure 2: Divide an attributed graph $\mathcal{G}$ into a bipartite graph $\mathcal{G}_\mathcal{A}$ and an unattributed graph $\mathcal{G}_\mathcal{S}$.

information are disentangled in graph embeddings (*i.e.* independence and interpretability holds), the elements corresponding to the fixed factor should possess lower values in $\Delta z_\mathcal{G}$, which makes it easier to predict $y_\mathcal{G}$ with a low capacity classifier (*e.g.* linear classifier) upon $\Delta z_\mathcal{G}$. Based on this fact, we employ the prediction accuracy of $y_\mathcal{G}$ on a set of graphs as the structure-attribute disentanglement metric (*SAD-Metric*). The whole evaluation procedure is summarized in Algorithm 1.

In order to measure the extent of structure-attribute disentanglement in node embeddings, we further design a node-centric metric, named as *node-SAD-Metric*. The detailed definition and experimental results for this metric are presented in Sec. E.

### 3.4 THEORETICAL ANALYSIS

In this section, we theoretically illustrate that the disentanglement of structure and attribute representation is able to ease the burden of model optimization by shrinking the solution space.

For an attributed graph $\mathcal{G} = (\mathcal{V}, \mathcal{E}, \mathcal{A})$, we can regard each type of node attribute as an attribute node, which transforms graph $\mathcal{G}$ into another form, $\mathcal{G} = (\mathcal{V}, \mathcal{V}_\mathcal{A}, \mathcal{E}, \mathcal{E}_\mathcal{A})$ ($\mathcal{V}_\mathcal{A}$: the set of all attribute nodes, $\mathcal{E}_\mathcal{A}$: the edges connecting normal and attribute nodes), as shown in Fig. 2(a). This graph can be divided into two parts: (1) a bipartite graph reflecting attribute information, $\mathcal{G}_\mathcal{A} = (\mathcal{V}, \mathcal{V}_\mathcal{A}, \mathcal{E}_\mathcal{A})$ (Fig. 2(b)) and (2) an unattributed graph depicting graph structure, $\mathcal{G}_\mathcal{S} = (\mathcal{V}, \mathcal{E})$ (Fig. 2(c)).

We first give the definitions of topological space and the spaces for graph $\mathcal{G}_\mathcal{A}$ and $\mathcal{G}_\mathcal{S}$.

**Definition 1.** *A topological space* $\mathcal{T} = \big(X, \mathcal{N}(x)\big)$ *is composed of a set* $X$ *and a neighborhood function* $\mathcal{N}(x)$ *mapping each* $x \in X$ *to a subset of* $X$.

**Definition 2.** *The topological space for graph* $\mathcal{G}_\mathcal{A}$ *is* $\mathcal{T}_\mathcal{A} = \big(\mathcal{V} \cup \mathcal{V}_\mathcal{A}, \mathcal{N}_\mathcal{A}(v)\big)$, *and the topological space for graph* $\mathcal{G}_\mathcal{S}$ *is* $\mathcal{T}_\mathcal{S} = \big(\mathcal{V}, \mathcal{N}_\mathcal{S}(v)\big)$.

We consider two ways of graph embedding. The first way embeds graph $\mathcal{G}_\mathcal{A}$ and $\mathcal{G}_\mathcal{S}$ to a common embedding space, *i.e.* learning a function $f : \mathcal{T}_\mathcal{A} \times \mathcal{T}_\mathcal{S} \rightarrow Z$, while the second way embeds two graphs to separate embedding spaces, *i.e.* learning a function $\tilde{f} : \mathcal{T}_\mathcal{A} \times \mathcal{T}_\mathcal{S} \rightarrow Z_\mathcal{A} \times Z_\mathcal{S}$. When the dimension of embedding space is identical in these two ways, we propose that the dimension of the solution space of function $\tilde{f}$ is smaller.

**Proposition 1.** *If it holds that* $\dim(Z) = \dim(Z_\mathcal{A}) + \dim(Z_\mathcal{S})$, *we have* $\dim(\tilde{f}) \leqslant \dim(f)$.

The detailed proof of this proposition is provided in Sec. A. Proposition 1 tells that the structure-attribute disentanglement of graph representation narrows the solution space of the model, which enables us to train the graph encoder more effectively.

## 4 RELATED WORK

**Graph Representation Learning.** The early efforts towards learning low-dimensional embeddings of nodes/graphs focused on optimizing the objectives induced by random walk statistics (Perozzi et al., 2014; Tang et al., 2015; Grover & Leskovec, 2016) or matrix factorization (Cao et al., 2015; Wang et al., 2016). By contrast, the Graph Neural Networks (GNNs) (Scarselli et al., 2008) derive embedding vectors via a neighborhood aggregation mechanism. Gilmer et al. (2017) suggested that most GNNs perform a *Message Passing* and a *Readout* phase, and different techniques (Kipf & Welling, 2017; Hamilton et al., 2017; Velickovic et al., 2018; Ying et al., 2018; Zhang et al., 2018; Xu et al., 2019) have been explored to enhance the effectiveness of these two phases. Un-

like these methods which mix the information of graph structure and node attributes into a single representation, our approach aims to disentangle these two factors in node/graph representations.

**Learning Disentangled Representations.** A disentangled representation is expected to separate the distinct and informative *factors of variation* in the data (Bengio et al., 2013). Some previous works sought to achieve this goal under the guidance of weak supervision (Hinton et al., 2011; Kulkarni et al., 2015; Siddharth et al., 2017). On another line of research, representation disentanglement is pursued by various unsupervised/self-supervised techniques (Chen et al., 2016; Higgins et al., 2017; Kim & Mnih, 2018; Chen et al., 2018). For graph-structured data, a recent work (Ma et al., 2019) disentangled the latent factors of graph structure via a neighborhood routing mechanism. The proposed structure-attribute disentanglement is orthogonal to their work and also more general.

**Measuring Disentangled Representations.** The quantitative measurement of representation disentanglement is essential to compare different disentangling algorithms. Given the true factors of variation, various disentanglement metrics have been designed based on classifier (Karaletsos et al., 2016; Higgins et al., 2017; Kim & Mnih, 2018), mutual information estimation (Chen et al., 2018; Ridgeway & Mozer, 2018) or distribution entropy (Eastwood & Williams, 2018). The proposed SAD-Metric follows the embedding-based evaluation protocol as previous works, while it is novel on its data processing scheme which is tailored for graph-structured data.

## 5 EXPERIMENTS

### 5.1 EXPERIMENTAL SETUP

**Model configurations.** For all approaches evaluated in Secs. 5.2 and 5.3, we equip them with a GCN (Kipf & Welling, 2017) (hidden units' dimension: 300, readout function: mean pooling) with two/three layers for node/graph classification, respectively. The performance on other GNNs, *i.e.* GraphSAGE (Hamilton et al., 2017), GAT (Velickovic et al., 2018) and GIN (Xu et al., 2019), is reported in Sec. 5.5. For the Input-SAD model, a two-layer fully-connected network (hidden units' dimension: 300, activation function: ReLU) is adopted to encode the attribute component of graph. For the Embed-SAD model, the discriminator of the NCE estimator is built as an encoder-and-dot architecture: $T(\mathbf{x}_1, \mathbf{x}_2) = f(\mathbf{x}_1)^{\mathrm{T}} f(\mathbf{x}_2)$, where $f(\cdot)$ is modeled by two linear layers and a ReLU nonlinearity in between, and it projects the original feature vector to an inner-product space.

**Training details.** In node classification experiments, an Adam optimizer (Kingma & Ba, 2015) (learning rate: $1 \times 10^{-3}$) is employed to train the model for 1000 epochs, and, for graph classification tasks, we use an Adam optimizer (learning rate: $1 \times 10^{-3}$, batch size: 32) to perform optimization for 100 epochs. For the negative sampling in NCE estimator (Eq. 3), we utilize all the nodes other than the selected one as negative samples (*i.e.* $K = |\mathcal{V}| - 1$), while, on three large networks (*i.e.* PubMed, Coauthor-CS and Coauthor-Physics), 3000 nodes serve as negative samples. Unless otherwise stated, trade-off parameters $\lambda_1$ and $\lambda_2$ are set as 1 and 0.1 (sensitivity analysis is in Sec. 5.5), which is determined by the grid search on the validation sets of three citation networks.

**Performance comparisons.** We combine the proposed Input-SAD and Embed-SAD model with four kinds of GNNs (GCN, GraphSAGE, GAT and GIN) to verify their effectiveness. Furthermore, we compare our method with five existing approaches that seek to promote GNNs' representation learning capability, *i.e.* Multi-task (Tran, 2018), GraphMix (Verma et al., 2019), DropEdge (Rong et al., 2019), DisenGNN (Ma et al., 2019) and InitRes (Chen et al., 2020) (detailed settings are in Sec. B). Among which, Multi-task and DisenGNN relate to our method: the former performs link prediction and node/graph classification simultaneously; the latter disentangles node representations based on different generative causes of edges.

### 5.2 EXPERIMENTS OF NODE CLASSIFICATION

**Datasets.** We employ three citation networks (*i.e.* Cora, CiteSeer and PubMed (Sen et al., 2008)) and two larger coauthor networks (*i.e.* Coauthor-CS and Coauthor-Physics (Shchur et al., 2018)) for semi-supervised node classification (20 labeled nodes per category). The node attributes of three citation networks are the bag-of-words representation of the document, and the nodes in two coauthor networks are featured by the paper keywords for each author's papers. Edge attributes are not included in these datasets. The details about dataset statistics, dataset split and evaluation protocol are provided in Sec. C.

Table 1: Node classification accuracy (mean $\pm$ std %) on citation and coauthor networks.

| Method | Cora | CiteSeer | PubMed | Coauthor-CS | Coauthor-Physics |
|---|---|---|---|---|---|
| GCN-baseline (Kipf & Welling, 2017) | $81.23 \pm 0.62$ | $70.31 \pm 0.57$ | $78.50 \pm 0.47$ | $90.98 \pm 0.55$ | $93.16 \pm 0.67$ |
| Multi-task (Tran, 2018) | $81.68 \pm 0.92$ | $70.46 \pm 0.65$ | $78.81 \pm 0.74$ | $91.08 \pm 0.41$ | $93.69 \pm 0.67$ |
| GraphMix (Verma et al., 2019) | $82.05 \pm 0.67$ | $\mathbf{71.50} \pm 0.54$ | $78.92 \pm 1.27$ | $91.65 \pm 0.53$ | $93.59 \pm 0.74$ |
| DropEdge (Rong et al., 2019) | $81.47 \pm 0.96$ | $70.70 \pm 0.61$ | $77.27 \pm 0.69$ | $89.56 \pm 0.55$ | $91.81 \pm 0.92$ |
| DisenGNN (Ma et al., 2019) | $81.95 \pm 0.66$ | $70.81 \pm 1.30$ | $78.19 \pm 0.53$ | $90.83 \pm 0.69$ | $93.55 \pm 0.73$ |
| InitRes (Chen et al., 2020) | $81.51 \pm 0.80$ | $70.47 \pm 0.56$ | $78.60 \pm 0.50$ | $91.70 \pm 0.52$ | $93.91 \pm 0.41$ |
| Input-SAD | $56.95 \pm 1.25$ | $53.97 \pm 1.19$ | $70.96 \pm 0.52$ | $87.35 \pm 0.82$ | $89.79 \pm 0.79$ |
| Embed-SAD (w/o $\mathcal{L}_{\mathrm{recon}}$) | $81.92 \pm 0.70$ | $70.38 \pm 0.98$ | $78.32 \pm 0.63$ | $91.57 \pm 0.40$ | $93.55 \pm 0.64$ |
| Embed-SAD (w/o $\mathcal{I}_{\mathcal{S},\mathcal{A}}$) | $81.41 \pm 0.69$ | $71.02 \pm 0.88$ | $78.92 \pm 0.43$ | $91.08 \pm 0.57$ | $93.34 \pm 0.50$ |
| Embed-SAD | $\mathbf{83.01} \pm 0.42$ | $71.23 \pm 1.22$ | $\mathbf{79.56} \pm 1.00$ | $\mathbf{91.85} \pm 0.79$ | $\mathbf{94.03} \pm 0.58$ |

Table 2: Test ROC-AUC (mean $\pm$ std %) on molecular graph classification benchmarks.

| Method | BBBP | Tox21 | ToxCast | SIDER | ClinTox | MUV | HIV | BACE | Avg |
|---|---|---|---|---|---|---|---|---|---|
| GCN-baseline (2017) | $68.1 \pm 0.3$ | $74.0 \pm 0.1$ | $63.9 \pm 0.1$ | $61.2 \pm 0.3$ | $61.6 \pm 0.6$ | $74.8 \pm 0.2$ | $73.3 \pm 0.2$ | $79.3 \pm 0.1$ | 69.5 |
| Multi-task (2018) | $71.3 \pm 0.1$ | $73.6 \pm 0.2$ | $63.6 \pm 0.1$ | $61.8 \pm 0.4$ | $60.4 \pm 2.1$ | $75.9 \pm 1.3$ | $76.4 \pm 1.0$ | $77.4 \pm 0.5$ | 70.1 |
| GraphMix (2019) | $67.5 \pm 0.3$ | $73.7 \pm 0.2$ | $63.2 \pm 0.1$ | $61.5 \pm 0.5$ | $62.5 \pm 1.6$ | $73.3 \pm 0.5$ | $73.1 \pm 0.4$ | $79.3 \pm 0.7$ | 69.3 |
| DropEdge (2019) | $70.7 \pm 0.2$ | $73.7 \pm 0.4$ | $63.3 \pm 0.2$ | $63.8 \pm 0.3$ | $66.5 \pm 3.8$ | $74.5 \pm 0.4$ | $75.4 \pm 0.6$ | $79.1 \pm 0.2$ | 70.9 |
| DisenGNN (2019) | $69.3 \pm 0.4$ | $74.8 \pm 0.1$ | $64.2 \pm 0.2$ | $61.4 \pm 0.1$ | $\mathbf{75.3} \pm 0.3$ | $74.2 \pm 1.1$ | $75.3 \pm 0.5$ | $81.7 \pm 0.3$ | 72.0 |
| InitRes (2020) | $71.9 \pm 0.9$ | $74.5 \pm 0.4$ | $63.9 \pm 0.3$ | $63.4 \pm 0.5$ | $63.0 \pm 1.7$ | $71.8 \pm 0.7$ | $76.6 \pm 0.4$ | $75.1 \pm 0.5$ | 70.0 |
| Input-SAD | $69.9 \pm 0.7$ | $74.3 \pm 0.2$ | $63.4 \pm 0.3$ | $\mathbf{63.9} \pm 0.4$ | $72.6 \pm 1.2$ | $74.0 \pm 0.6$ | $74.2 \pm 0.4$ | $\mathbf{82.1} \pm 0.6$ | 71.8 |
| Embed-SAD (w/o $\mathcal{L}_{\mathrm{recon}}$) | $71.4 \pm 0.3$ | $75.5 \pm 0.3$ | $64.1 \pm 0.2$ | $63.2 \pm 0.5$ | $64.5 \pm 0.4$ | $76.0 \pm 0.6$ | $75.4 \pm 0.1$ | $80.7 \pm 0.7$ | 71.4 |
| Embed-SAD (w/o $\mathcal{I}_{\mathcal{S},\mathcal{A}}$) | $72.3 \pm 0.2$ | $75.0 \pm 0.3$ | $64.1 \pm 0.1$ | $63.3 \pm 0.3$ | $67.2 \pm 0.5$ | $74.8 \pm 0.7$ | $75.1 \pm 0.3$ | $80.4 \pm 0.4$ | 71.5 |
| Embed-SAD | $\mathbf{73.0} \pm 0.2$ | $\mathbf{75.6} \pm 0.2$ | $\mathbf{64.4} \pm 0.3$ | $63.3 \pm 0.2$ | $74.1 \pm 1.1$ | $\mathbf{76.1} \pm 0.2$ | $\mathbf{76.9} \pm 0.4$ | $81.3 \pm 0.1$ | $\mathbf{73.1}$ |

Table 3: SAD-Metric score (mean $\pm$ std %) on molecular graph datasets.

| Method | BBBP | Tox21 | ToxCast | SIDER | ClinTox | MUV | HIV | BACE | Avg |
|---|---|---|---|---|---|---|---|---|---|
| random-GCN (2017) | $70.5 \pm 0.4$ | $75.5 \pm 0.3$ | $73.9 \pm 0.2$ | $82.1 \pm 0.4$ | $69.9 \pm 1.5$ | $76.5 \pm 0.8$ | $75.8 \pm 0.6$ | $82.2 \pm 0.3$ | 75.6 |
| GCN-baseline (2017) | $82.5 \pm 0.2$ | $88.9 \pm 0.1$ | $90.1 \pm 0.1$ | $94.9 \pm 0.3$ | $86.0 \pm 0.8$ | $92.4 \pm 0.4$ | $83.1 \pm 0.2$ | $94.7 \pm 0.5$ | 89.1 |
| DisenGNN (2019) | $92.3 \pm 0.8$ | $88.4 \pm 0.2$ | $93.4 \pm 0.4$ | $95.6 \pm 0.7$ | $95.0 \pm 0.3$ | $93.8 \pm 0.2$ | $93.2 \pm 0.4$ | $93.9 \pm 0.3$ | 93.2 |
| Input-SAD | $93.3 \pm 0.7$ | $94.5 \pm 0.3$ | $96.0 \pm 0.1$ | $93.6 \pm 0.5$ | $\mathbf{96.4} \pm 0.4$ | $93.2 \pm 0.6$ | $94.1 \pm 0.4$ | $96.3 \pm 0.4$ | 94.7 |
| Embed-SAD | $\mathbf{95.9} \pm 0.2$ | $\mathbf{98.7} \pm 0.6$ | $\mathbf{98.6} \pm 0.9$ | $\mathbf{97.3} \pm 0.1$ | $95.0 \pm 0.6$ | $\mathbf{98.5} \pm 0.1$ | $\mathbf{95.2} \pm 0.2$ | $\mathbf{97.8} \pm 0.6$ | $\mathbf{97.1}$ |

**Results.** In Tab. 1, we report the performance of different methods on five standard node classification benchmarks. Based on a two-layer GCN model, the proposed Embed-SAD achieves the highest test accuracy on four of five tasks among all approaches. The performance of Input-SAD is inferior on these tasks, which, we think, is mainly ascribed to its failure in fully capturing the structure and attribute information of each node.

## 5.3 EXPERIMENTS OF GRAPH CLASSIFICATION

**Datasets.** For graph classification, we adopt eight single-task/multi-task classification datasets in MoleculeNet (Wu et al., 2018), and, following previous findings (Chen et al., 2012), a scaffold split scheme is utilized for dataset split. For each molecular graph sample, the type and chirality of atom serve as the node attributes, and the type and direction of bond constitute the edge attributes. For evaluation, five independent runs with distinct random seeds are conducted, and the average performance is reported. More dataset statistics are in Sec. C.

**Results.** Tab. 2 shows the comparisons between our methods and existing techniques. Embed-SAD outperforms DisenGNN, a closely related work, on six of eight datasets, and a $1.1\%$ performance gain is achieved in terms of average ROC-AUC, which illustrates the effectiveness of structure-attribute disentanglement. Input-SAD performs well on this graph classification benchmark and obtains superior performance on SIDER and BACE datasets, which, we think, is because the graph pooling operation is able to supplement the missing structure or attribute information of each node from other nodes.

## 5.4 QUANTITATIVE EVALUATION OF STRUCTURE-ATTRIBUTE DISENTANGLEMENT

**Evaluation details.** We employ the same datasets and dataset split scheme as in Sec. 5.3. We use a GCN model with random parameters (random-GCN) and a GCN model pre-trained by graph classification (GCN-baseline) as baseline models. Three disentanglement models (DisenGNN, Input-SAD and Embed-SAD) also establish appropriate priors by performing graph classification on the training set. A linear binary classifier is built upon the model to be evaluated, and the classifier is

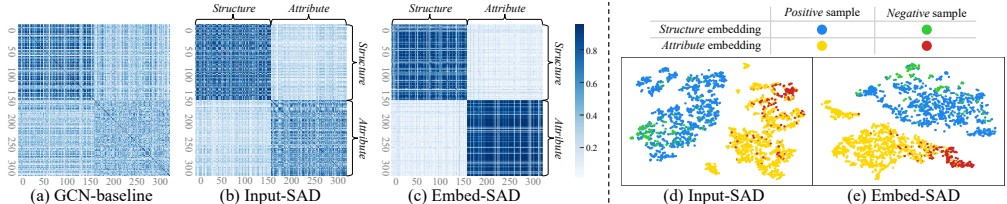

Figure 3: (a) Performance comparisons on four GNNs. (b)&(c) Sensitivity analysis of $\lambda_1$ and $\lambda_2$.

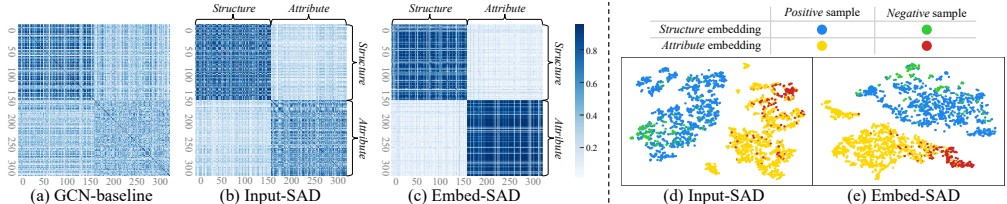

Figure 4: Visualization of correlation matrix and embedding distributions on BBBP's training set.

trained with the graphs in the training set and evaluated by test graphs. All results are averaged over five independent runs.

**Results.** We report the SAD-Metric score of five methods in Tab. 3. The poor performance of random-GCN shows that it can hardly disentangle the structure and attribute information of a graph, and the graph classification pre-training endows the GCN-baseline model with better disentanglement capability. Compared to DisenGNN, the proposed Input-SAD and Embed-SAD model better disentangle graph's structure and attribute information. The Embed-SAD model achieves the best disentanglement of graph representations on seven of eight datasets.

## 5.5 ANALYSIS

**Effect of two objectives on Embed-SAD.** In Tabs. 1 and 2, we analyze the effect of reconstruction loss and structure-attribute mutual information minimization. When removing any of these two objectives, the model's performance is impaired, which demonstrates their complementary relation.

**Results of different GNNs.** In Fig. 3(a), we combine three representation disentanglement techniques with four kinds of GNNs, and the average test ROC-AUC over eight datasets of MoleculeNet is plotted (detailed results are in Sec. D). Embed-SAD performs best on all four configurations, and the performance of Input-SAD and DisenGNN is comparable with each other.

**Sensitivity of trade-off parameters** $\lambda_1$, $\lambda_2$**.** Figs. 3(b), (c) show the performance of Embed-SAD on citation networks using different trade-off weights. We can observe that stable performance gain is obtained when the values of $\lambda_1$ and $\lambda_2$ are around 1 and 0.1, respectively.

**Visualization.** In Figs. 4(a), (b), (c), we visualize the absolute values of the correlations between the learnt graph embedding's elements on BBBP dataset. Among three models, Embed-SAD suppresses the correlation between structure and attribute embedding to the greatest extent. We further visualize the embedding distributions using t-SNE (Maaten & Hinton, 2008) in Figs. 4(d), (e). Both Input-SAD and Embed-SAD separate two kinds of embeddings into distinct spaces, and attribute embeddings possess stronger semantic discriminability compared to the structure counterparts.

## 6 CONCLUSIONS AND FUTURE WORK

We study node/graph representation learning with Structure-Attribute Disentanglement (GraphSAD) in both the input and the embedding space. We further design a quantitative metric to measure such disentanglement. On node and graph classification benchmark datasets, we empirically verify our method's superior performance over existing techniques.

Our future explorations will involve improving the learning manners for structure-attribute disentangled representations, evaluating the proposed models on more tasks (*e.g.* regression-based tasks) and disentangling graphs in other ways.

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

## A   PROOF OF PROPOSITION 1

Considering the search of embedding function as a combinatorial search problem (Bruynooghe, 1981; Chen et al., 2001; Kotthoff, 2016), we can derive the dimension of the solution space for two types of embedding functions, *i.e.* $f : \mathcal{T}_\mathcal{A} \times \mathcal{T}_\mathcal{S} \to Z$ and $\tilde{f} : \mathcal{T}_\mathcal{A} \times \mathcal{T}_\mathcal{S} \to Z_\mathcal{A} \times Z_\mathcal{S}$, as follows:

$$\dim(f) = \binom{\dim(\mathcal{T}_\mathcal{A}) + \dim(\mathcal{T}_\mathcal{S})}{\dim(Z)}, \tag{13}$$

$$\dim(\tilde{f}) = \binom{\dim(\mathcal{T}_\mathcal{A})}{\dim(Z_\mathcal{A})} \cdot \binom{\dim(\mathcal{T}_\mathcal{S})}{\dim(Z_\mathcal{S})}. \tag{14}$$

Under the condition that $\dim(Z) = \dim(Z_\mathcal{A}) + \dim(Z_\mathcal{S})$, the basic property of combinations gives the following inequality:

$$\binom{\dim(\mathcal{T}_\mathcal{A})}{\dim(Z_\mathcal{A})} \cdot \binom{\dim(\mathcal{T}_\mathcal{S})}{\dim(Z_\mathcal{S})} \leqslant \binom{\dim(\mathcal{T}_\mathcal{A}) + \dim(\mathcal{T}_\mathcal{S})}{\dim(Z)}, \tag{15}$$

which deduces that:

$$\dim(\tilde{f}) \leqslant \dim(f). \tag{16}$$

This conclusion shows that the solution space of the graph embedding function with structure-attribute disentanglement is narrower than that of the entangled counterpart.

## B   MORE IMPLEMENTATION DETAILS

**Modeling conditional distribution for edge reconstruction.** For the graphs that only provide the existence of edges (*i.e.* $N_t = 1$), we utilize an inner-product decoder for edge prediction:

$$p(y = 0 | z_{u,\mathcal{S}}, z_{v,\mathcal{S}}) = 1 - \sigma(z_{u,\mathcal{S}}^\mathrm{T} z_{v,\mathcal{S}}), \quad p(y = 1 | z_{u,\mathcal{S}}, z_{v,\mathcal{S}}) = \sigma(z_{u,\mathcal{S}}^\mathrm{T} z_{v,\mathcal{S}}), \tag{17}$$

where $\sigma(\cdot)$ denotes the sigmoid function.

For the graphs owning more than one type of edges (*i.e.* $N_t > 1$), we concatenate the embedding vectors of two nodes and use a linear classifier to predict the type of the edge between them:

$$p(y | z_{u,\mathcal{S}}, z_{v,\mathcal{S}}) = F\big([z_{u,\mathcal{S}}, z_{v,\mathcal{S}}]\big), \tag{18}$$

where $F : \mathbb{R}^{2\delta} \to \mathbb{R}^{N_t + 1}$ is the edge prediction function modeled by a linear classifier network.

**Attribute modification for SAD-Metric.** To modify the attribute of a graph, we randomly select a node in this graph and alter its attribute. Specifically, for a graph sample (*i.e.* molecule) in the datasets of MoleculeNet, we modify its attribute by randomly selecting an atom and resetting the atom's type and chirality to other valid values. Such technique can be applied to any graph dataset in which node attributes are discretely represented.

**Detailed settings for compared methods.** In the experiments, we compare our method with five existing techniques that aim to improve graph representation learning, *i.e.* Multi-task (Tran, 2018), GraphMix (Verma et al., 2019), DropEdge (Rong et al., 2019), DisenGNN (Ma et al., 2019) and InitRes (Chen et al., 2020). The detailed settings of these approaches are as follows:

- *Multi-task.* Node embeddings are employed for both node classification and link prediction, and the loss term for link prediction possesses the weight of 0.1.
- *GraphMix.* In this approach, parameter $\alpha$ controls the Beta distribution from which mixup ratios are sampled, and we fix this parameter as $1.0$ in all experiments.
- *DropEdge.* During training, $10\%$ edges of each graph are randomly dropped to mitigate the over-fitting and over-smoothing problems.
- *DisenGNN.* For each GNN layer, the feature vector of each node is divided into 5 channels, and 5 iterations of neighborhood routing are performed.
- *InitRes.* This method constructs a residual connection from the initial node representations to each GNN layer. We set the residual ratio as 0.2, such that the final representation of each node retains at least $20\%$ of the input feature.

## C   MORE EXPERIMENTAL DETAILS

Table 4: Dataset statistics for citation and coauthor networks.

| Dataset | # Nodes | # Edges | # Features | # Classes | # Training | # Validation | # Test |
|---------|---------|---------|-----------|-----------|------------|--------------|--------|
| Cora | 2,708 | 5,429 | 1,433 | 7 | 140 | 500 | 1,000 |
| CiteSeer | 3,327 | 4,732 | 3,703 | 6 | 120 | 500 | 1,000 |
| PubMed | 19,717 | 44,338 | 500 | 3 | 60 | 500 | 1,000 |
| Coauthor-CS | 18,333 | 81,894 | 6,805 | 15 | 300 | 450 | 17,583 |
| Coauthor-Physics | 34,493 | 247,962 | 8,415 | 5 | 100 | 150 | 34,243 |

Table 5: Dataset statistics for molecule datasets.

| Dataset | BBBP | Tox21 | ToxCast | SIDER | ClinTox | MUV | HIV | BACE |
|---------|------|-------|---------|-------|---------|-----|-----|------|
| # Molecules | 2,039 | 7,831 | 8,575 | 1,427 | 1,478 | 93,087 | 41,127 | 1,513 |
| # Tasks | 1 | 12 | 617 | 27 | 2 | 17 | 1 | 1 |
| Avg Nodes | 24.06 | 18.57 | 18.78 | 33.64 | 26.16 | 24.23 | 25.51 | 34.09 |
| Avg Edges | 25.95 | 19.29 | 19.26 | 35.36 | 27.88 | 26.28 | 27.47 | 36.86 |

**Citation networks.** In Tab. 4, we provide the detailed information about three citation networks, in which 20 labeled nodes per category are used for training, and there are 500 and 1000 nodes for validation and test, respectively. We adopt the standard dataset split proposed in Yang et al. (2016) for all experiments on citation networks. All the reported results are averaged over 100 independent runs using different random seeds.

**Coauthor networks.** The statistics of two coauthor networks are listed in Tab. 4. Following Shchur et al. (2018), we use 20 labeled nodes per category as the training set, 30 nodes per category as the validation set, and the rest as the test set. The reported accuracy is averaged over 30 random train/validation/test splits, and, for each split, 50 independent runs are performed.

**Molecule datasets.** In Tab. 5, we present the detailed statistics of eight molecule datasets in MoleculeNet (Wu et al., 2018). On each dataset, a model intends to predict one or multiple properties of various molecules, where the prediction of each property is a binary classification task.

## D   MORE RESULTS ON MOLECULENET

In Tabs. 6, 7 and 8, we combine different techniques with three GNNs (*i.e.* GraphSAGE (Hamilton et al., 2017), GAT (Velickovic et al., 2018) and GIN (Xu et al., 2019)) and evaluate various models' performance on MoleculeNet, in which we set the number of attention heads as 5 for GAT. For all the three experiments, the Embed-SAD model outperforms other methods in terms of average test ROC-AUC. The Input-SAD model achieves superior performance on ClinTox and BACE datasets, and its overall performance is comparable with DisenGNN.

## E   NODE-SAD-METRIC: NODE-CENTRIC STRUCTURE-ATTRIBUTE DISENTANGLEMENT METRIC

### E.1   DEFINITION

The *node-SAD-Metric* measures the extent of structure-attribute disentanglement in node embeddings. Similar with the graph-level SAD-Metric, this node-level disentanglement metric evaluate two properties of node embeddings, *i.e.* independence and interpretability (detailed definitions referring to Sec. 3.3). In this case, after structure and attribute modification, the nodes of a graph are classified into three types: (1) the nodes whose one-hop structure is modified, (2) the nodes whose attribute is modified and (3) the nodes without one-hop structure or attribute modification. In practice, we first randomly drop 20% edges in a graph and then randomly modify the attributes of 20% of the rest nodes whose one-hop structure have not been modified. Similar as in SAD-Metric (Sec. 3.3), we employ the absolute difference between the node embeddings before and after the above modification to perform a three-way classification. Upon on such embedding difference, a

Table 6: Test ROC-AUC (mean ± std %) on MoleculeNet using the GraphSAGE architecture.

| Method | BBBP | Tox21 | ToxCast | SIDER | ClinTox | MUV | HIV | BACE | Avg |
|---|---|---|---|---|---|---|---|---|---|
| GraphSAGE-baseline (2017) | 70.9 ± 1.0 | 74.3 ± 0.1 | 63.9 ± 0.2 | 61.5 ± 0.2 | 65.4 ± 0.2 | 78.2 ± 0.6 | 74.6 ± 0.5 | 76.0 ± 0.8 | 70.6 |
| Multi-task (2018) | 71.3 ± 0.4 | 73.7 ± 0.1 | 62.8 ± 0.2 | 63.2 ± 0.5 | 64.8 ± 1.6 | **79.7** ± 0.3 | 73.5 ± 0.4 | 67.6 ± 0.6 | 69.6 |
| GraphMix (2019) | 70.9 ± 0.5 | 73.8 ± 0.3 | 63.7 ± 0.3 | 62.5 ± 0.9 | 63.5 ± 0.9 | 73.9 ± 0.4 | 75.4 ± 0.6 | 77.5 ± 0.7 | 70.1 |
| DropEdge (2019) | 71.3 ± 0.3 | 74.7 ± 0.1 | 62.6 ± 0.3 | 63.1 ± 0.3 | 66.1 ± 2.4 | 76.2 ± 0.3 | 75.9 ± 1.4 | 76.7 ± 0.3 | 70.8 |
| DisenGNN (2019) | 72.4 ± 0.6 | **75.2** ± 0.4 | 63.7 ± 0.3 | 61.5 ± 0.5 | 69.6 ± 2.6 | 75.7 ± 0.7 | 73.4 ± 0.7 | 80.2 ± 0.1 | 71.5 |
| InitRes (2020) | 71.5 ± 1.3 | 75.0 ± 0.1 | 63.4 ± 0.1 | 63.7 ± 0.3 | 64.6 ± 1.5 | 73.7 ± 0.3 | 75.0 ± 0.6 | 75.3 ± 0.8 | 70.3 |
| Input-SAD | 69.2 ± 0.7 | 73.4 ± 0.4 | 62.8 ± 0.2 | 62.3 ± 0.5 | 70.4 ± 0.5 | 75.2 ± 0.6 | 74.2 ± 0.8 | **82.5** ± 0.9 | 71.3 |
| Embed-SAD | **74.4** ± 0.3 | 74.5 ± 0.3 | **64.3** ± 0.1 | **63.9** ± 0.3 | 70.7 ± 0.4 | 78.4 ± 0.2 | **76.7** ± 0.5 | 78.7 ± 0.7 | **72.7** |

Table 7: Test ROC-AUC (mean ± std %) on MoleculeNet using the GAT architecture.

| Method | BBBP | Tox21 | ToxCast | SIDER | ClinTox | MUV | HIV | BACE | Avg |
|---|---|---|---|---|---|---|---|---|---|
| GAT-baseline (2018) | 70.0 ± 0.2 | 67.0 ± 0.1 | 61.6 ± 0.3 | 62.6 ± 0.6 | 64.6 ± 1.5 | 69.5 ± 0.5 | 65.6 ± 0.1 | 72.3 ± 1.3 | 66.6 |
| Multi-task (2018) | 68.2 ± 0.3 | 67.2 ± 0.9 | 58.2 ± 0.5 | 60.1 ± 0.2 | 64.3 ± 0.6 | 63.7 ± 4.4 | 66.5 ± 0.9 | 69.0 ± 2.3 | 64.6 |
| GraphMix (2019) | 68.7 ± 0.3 | 65.9 ± 0.7 | 59.5 ± 0.7 | 60.8 ± 0.4 | 64.5 ± 0.7 | 67.9 ± 1.0 | 66.3 ± 1.1 | 71.9 ± 1.0 | 65.7 |
| DropEdge (2019) | 67.7 ± 0.3 | 68.4 ± 0.3 | 61.7 ± 0.4 | 62.3 ± 0.3 | 67.7 ± 2.5 | 65.5 ± 0.5 | 67.8 ± 0.3 | 72.3 ± 0.4 | 66.7 |
| DisenGNN (2019) | 69.0 ± 0.5 | 66.3 ± 0.5 | 60.9 ± 0.1 | 62.4 ± 0.5 | 63.1 ± 0.4 | 70.1 ± 0.7 | **71.6** ± 0.9 | 70.3 ± 0.5 | 66.7 |
| InitRes (2020) | 70.3 ± 0.3 | 70.1 ± 0.7 | **63.0** ± 0.1 | 60.9 ± 0.3 | 69.3 ± 2.1 | 66.5 ± 0.7 | 71.5 ± 1.1 | 71.8 ± 0.3 | 67.9 |
| Input-SAD | 68.0 ± 1.1 | 70.1 ± 0.2 | 62.0 ± 0.3 | 59.9 ± 0.4 | **71.8** ± 0.4 | 68.0 ± 0.1 | 70.8 ± 0.3 | **74.4** ± 0.6 | 68.1 |
| Embed-SAD | **71.5** ± 0.6 | 70.9 ± 0.3 | 60.8 ± 0.2 | **62.9** ± 0.1 | 69.5 ± 1.6 | **70.3** ± 1.1 | 68.7 ± 1.2 | 72.6 ± 0.6 | **68.4** |

Table 8: Test ROC-AUC (mean ± std %) on MoleculeNet using the GIN architecture.

| Method | BBBP | Tox21 | ToxCast | SIDER | ClinTox | MUV | HIV | BACE | Avg |
|---|---|---|---|---|---|---|---|---|---|
| GIN-baseline (2019) | 72.1 ± 0.2 | 75.3 ± 0.6 | 62.4 ± 0.2 | 61.3 ± 0.5 | 65.5 ± 0.6 | 75.6 ± 0.9 | 75.2 ± 0.7 | 79.9 ± 1.0 | 70.9 |
| Multi-task (2018) | 72.0 ± 1.4 | 74.9 ± 0.2 | **64.1** ± 0.2 | 61.7 ± 0.5 | 66.4 ± 1.0 | 76.4 ± 0.6 | 76.9 ± 0.3 | 77.0 ± 0.7 | 71.2 |
| GraphMix (2019) | 71.4 ± 0.7 | 74.9 ± 0.2 | 62.2 ± 0.1 | 62.3 ± 0.9 | 63.9 ± 0.1 | 77.5 ± 0.3 | 75.8 ± 0.2 | 79.9 ± 1.3 | 71.0 |
| DropEdge (2019) | 72.3 ± 1.1 | 74.9 ± 0.4 | 63.5 ± 0.1 | 61.8 ± 0.7 | 68.2 ± 1.5 | 73.9 ± 0.2 | 75.7 ± 0.6 | 80.6 ± 0.8 | 71.4 |
| DisenGNN (2019) | 71.8 ± 0.4 | 74.4 ± 0.3 | 64.0 ± 0.3 | 61.4 ± 0.3 | 72.1 ± 1.5 | 76.0 ± 0.8 | 76.4 ± 0.3 | 80.0 ± 1.1 | 72.0 |
| InitRes (2020) | 72.2 ± 0.8 | 75.8 ± 0.5 | 63.8 ± 1.1 | 61.8 ± 0.7 | 69.1 ± 0.3 | 72.0 ± 1.7 | 76.1 ± 1.7 | 80.1 ± 0.1 | 71.3 |
| Input-SAD | 69.7 ± 0.2 | 73.2 ± 0.3 | 63.2 ± 0.6 | 62.2 ± 0.4 | 75.1 ± 2.5 | 74.0 ± 0.6 | 76.1 ± 0.7 | **83.5** ± 1.4 | 72.1 |
| Embed-SAD | **73.6** ± 0.6 | **75.9** ± 0.3 | 63.5 ± 0.2 | **62.6** ± 0.1 | **78.3** ± 1.6 | **77.6** ± 1.1 | **77.5** ± 1.2 | 80.4 ± 0.6 | **73.7** |

Table 9: The node-SAD-Metric score (mean ± std %) on molecular graph datasets.

| Method | BBBP | Tox21 | ToxCast | SIDER | ClinTox | MUV | HIV | BACE | Avg |
|---|---|---|---|---|---|---|---|---|---|
| random-GCN (2017) | 77.4 ± 0.6 | 78.0 ± 0.6 | 78.6 ± 0.5 | 80.2 ± 0.7 | 78.1 ± 0.3 | 83.9 ± 0.8 | 81.1 ± 0.3 | 76.5 ± 1.0 | 79.2 |
| GCN-baseline (2017) | 89.7 ± 0.6 | 87.5 ± 0.5 | 90.6 ± 0.9 | 89.2 ± 0.8 | 86.9 ± 1.0 | 92.4 ± 0.3 | 92.2 ± 0.8 | 88.9 ± 0.7 | 89.7 |
| DisenGNN (2019) | 92.0 ± 0.9 | 91.2 ± 0.6 | 93.3 ± 0.3 | 92.5 ± 0.7 | 90.1 ± 0.5 | 94.3 ± 0.5 | 93.6 ± 0.4 | 91.3 ± 0.7 | 92.3 |
| Input-SAD | 93.4 ± 1.0 | 92.8 ± 0.6 | 95.8 ± 0.3 | 93.7 ± 0.7 | 93.3 ± 0.4 | **96.7** ± 0.8 | 96.4 ± 0.8 | 92.9 ± 0.7 | 94.4 |
| Embed-SAD | **95.8** ± 0.5 | **95.1** ± 0.6 | **97.2** ± 0.8 | **95.7** ± 0.5 | **95.4** ± 0.2 | 96.5 ± 0.3 | **97.0** ± 0.7 | **96.1** ± 1.1 | **96.1** |

linear classifier is trained to classify which type of modification is performed on a node, and the prediction accuracy of such node classification task serves as the node-SAD-Metric.

### E.2 EXPERIMENTAL RESULTS

**Setups.** As in Sec. 5.4, we use the MoleculeNet dataset (Wu et al., 2018) and the scaffold split scheme (Chen et al., 2012) for this experiment. Also, the settings of five studied models are identical with those in Sec. 5.4. A linear classifier is trained with the graphs in the training split and evaluated by the graphs in the test split. All results are averaged over five independent runs using different random seeds.

**Results.** Tab. 9 reports the structure-attribute disentanglement performance of five models in terms of node-SAD-Metric. The random-GCN baseline can poorly disentangle the structure and attribute information in node embedding, and GCN-baseline performs better by pre-training on graph classification task. Among three methods for disentangled representation learning, the Embed-SAD model achieves the best performance on seven of eight tasks.

