# OpenReview forum: "GraphSAD: Learning Graph Representations with Structure-Attribute Disentanglement"
_ICLR.cc/2021/Conference — Reject_

### Official Review · AnonReviewer4 · 2020-10-22
**Feel confused on the motivation**

**Rating:** 3
**Confidence:** 3

**Review:**

1. I am confused on the motivation part. Although the authors provide two examples (the prediction of a user’s social class mainly relies on his/her social network structure) to illustrate the benefits of disentangled representation for certain downstream tasks and I agree on the statement, it raises another question that what kind of representation, structure component or attribute component should be used for a specific task.
2. If we take a further through on the relationship between structure component and attribute component, it should be separated into three parts, a common part, structure-specific part and attribute-specific part. In my eyes, the authors put the common part and structure-specific part together. Why? What is the benefit? So, the name "structure/attribute component " is not accurate.
3. I do not see too much novelty in the technical part (edge reconstruction and mutual information minimizing). I am fine with this. But I do not see a deep insight of the problem-solving philosophy, either.
4. It is nice to see the performance gain in the experimental part. But  I want to know the authors use one type or two types of representation.

---

> ### Author Response · Authors · 2020-11-17
> **Official Response to AnonReviewer4**
>
> Thanks for your insightful comments!
>
> We respond to your questions as follows:
>
> Q1: How to determine whether structure or attribute representation should be used for a specific task?
>
> A1: Our method is motivated by the fact that different downstream tasks rely on different information in a graph, i.e. graph structure or node attribute, and thus we propose to disentangle these two types of information in node/graph embeddings.
>
> We would like to state that this motivation is not bothered by the selection between structure and attribute component for a specific task. Given a downstream task, we use a learnable task-specific classifier upon disentangled representations to judge which kinds of information is more important to this task. For example, if the prediction result of a specific category in the task is mainly derived by the structure embedding (i.e. the weights corresponding to structure embedding are higher in a fully-connected layer), we can deem that structure information is more critical for this task-specific prediction.
>
> Q2: Where is the common information between graph structure and node attributes encoded in the proposed model? Why not encoding such common information into an unique representation?
>
> A2: It is true that there exists the common information shared between graph structure and node attributes, but, in our perspective, it is better to encode such common information into both structure and attribute embedding, instead of using an unique embedding to represent it. The reasons are as follows: (1) Such unique embedding for common information lacks exact semantics (i.e. we cannot know what information it truly encodes), which impairs the interpretability of the whole model; (2) This common information can be sufficiently captured by structure and attribute embedding, which is illustrated by the non-zero structure-attribute mutual information (in Embed-SAD, mutual information $I_{S,A}$ is about 0.7~0.8 when the model converges). Considering these facts, in our method, we encode the common information between graph structure and node attributes into both structure and attribute embedding without disentangling it uniquely.
>
> Q3: The technical novelty of this work is limited.
>
> A3: To the best of our knowledge, this work is the first attempt towards structure-attribute disentangled graph representation learning, which mainly focuses on exploring this new problem with easy-to-understand and comprehensive (input-level, embedding-level and metric-level) techniques. More in-depth research with entirely innovative techniques on this topic will be the direction of our future work.
>
> Q4: In the experiment part, do one type or two types of representations are used?
>
> A4: In all experiments, both the Input-SAD and Embed-SAD model utilize the concatenation of structure and attribute embedding as the input of the following task-specific network (e.g. classifier), and the importance of these two types of representations is determined by the downstream task through learning task-specific model parameters.

---

> > ### Comment · AnonReviewer4 · 2020-11-17
> > **Response to Authors**
> >
> > Thanks for the response and clarification.
> > 1. With extra explanations, some of my concerns are addressed. But I am still stuck on the motivation part. The goal of this paper is to disentangle these two types (structure and attribute) of information in node/graph embeddings. However, the concatenation of structure and attribute embedding is used as the input of the following task-specific network. I do not understand the motivation or meaning of disentangling.
> > 2. In your framework, the common part only goes to the structural part, rather than both. Right?
> >
> > Here is my understanding. The authors aim to learn a new representation, which can explicitly be separated into structure and attribute embedding. However, the structure part contains structure-specific information and common information, while the attribute part only contains the attribute-specific information. In the experiments, the authors use both of them for the downstream task. This loses the meaning of disentangling. If the authors want to increase the performance, why not to learn the task-specific representation? With the label guidance, it should have a better performance than the structural-attribute embedding.

---

> > > ### Author Response · Authors · 2020-11-18
> > > **Official Response to AnonReviewer4**
> > >
> > > Thanks for your feedback!
> > >
> > > We further elucidate the motivation and respond to your questions as follows:
> > >
> > > Q1: What’s the motivation or meaning of concatenating the structure and attribute embedding and feeding the concatenated embedding to the task-specific network?
> > >
> > > A1: The motivations of disentangling two factors of variation (graph structure and node attributes) in node/graph representations and employing the concatenated representations of both factors for downstream task stem from the following considerations:
> > >
> > > (1) It is hard to heuristically judge which of these two factors is more important for a downstream task (such judgement is better to be done by a machine learning model upon structure-attribute concatenated representation), and, for many tasks, both factors can provide useful information (e.g. predicting the job of a social network user depends on both his/her profile information and social relationships).
> > >
> > > (2) Compared with using a single representation entangling the information of graph structure and node attributes, a structure-attribute disentangled representation can ease task-specific network’s burden of selecting task-relevant features from numerous highly-entangled feature elements and thus boost model’s performance. Also, such disentangled representation can enhance model’s interpretability by making it easier to judge which type of information, structure or attribute, a specific task more relies on.
> > >
> > > Q2: Whether the common information only goes to the structure embedding, rather than both kinds of embeddings?
> > >
> > > A2: We would like to clarify that the common information shared between graph structure and node attributes is encoded in both types of embeddings, instead of only in the structure one.
> > >
> > > For the Input-SAD model, the structure/attribute component of a graph can contain some information from the opposite part. In specific, the structure component, an unattributed graph, can imply some structure-derived attributes for each node (e.g. the degree of a node), and, similarly, the attribute component is able to imply some structural information (e.g. a student node is highly possible to interact with a teacher node). As a result, though these two components are separately encoded in Input-SAD, both the derived structure and attribute embedding contain the common information.
> > >
> > > For the Embed-SAD model, the common information is preserved in two kinds of embeddings by the adversarial learning of mutual information. Specifically, a discriminator is optimized to maximize the mutual information between structure and attribute embedding, while the GNN encoder intends to minimize it. When such adversarial learning reaches convergence, the structure- and attribute-specific information can be distilled into structure and attribute embedding respectively, and the common information is preserved in both types of embeddings (the structure-attribute mutual information $I_{S,A}$ is about 0.7~0.8 when the model converges).
> > >
> > > Q3: Using both kinds of embeddings lose the meaning of disentangling. Why not learn the task-specific representation directly?
> > >
> > > A3: As stated in the response ‘A1’ for the first question, considering the fact that the relative importance of structure and attribute information can hard be heuristically judged for each task, and both of the information can be useful for some tasks, we employ both the structure and attribute embedding for the downstream task.
> > >
> > > Also, as explained in the response ‘A1’, compared with the entangled task-specific representation, the structure-attribute disentangled representation eases the burden of selecting task-relevant features and enhances model’s interpretability. The experimental results on node and graph classification benchmarks (**Tabs. 1 and 2**) show the superiority of the disentangled-representation-based methods (e.g. Embed-SAD) against the task-specific methods with entangled representation (e.g. GCN-baseline).

---

### Official Review · AnonReviewer3 · 2020-10-26
**This paper describes a graph representation learning method which disentangles structure information and feature information, and combines them for downstream tasks. Generally speaking, I think this is a good paper. But some design choices and explanations are not very convincing.**

**Rating:** 6
**Confidence:** 4

**Review:**

This paper describes a graph representation learning method which disentangles structure information and feature information, and combines them for downstream tasks. Generally speaking, I think this is a good paper. But some design choices and explanations are not very convincing. Details are as follows.


[Pros]
(1)	This paper is well written and easy to follow.
(2)	The proposed problem is novel and the solution is intuitive.
(3)	Experiments are thorough.

[Cons]
(1)	Regarding the SAD-Metric, why do you choose the undirect scheme based on a classifier? The goal is to minimize the change of structure (attribute) part of the embedding when altering the attributes (edges) of the graph. Using a classifier is a undirect measure about this, and would lead to misunderstanding when the classifier treats some elements for the structure (attribute) part as those for the attribute (structure) part. Imagine that such elements are coincidentally correlated with the opposite part. In that case, the accuracy could also be high. I think here it could be better to design a measure directly to reflect the above goal. Another question is, why do you define the measure on the graph embedding level, rather than the node embedding level, where the node is the one affected by the modification. I think this scheme could lead to more sensitive measures.
(2)	The authors argue that the previous metrics for measuring disentangled representations are mainly designed for images, while the proposed SAD-Metric is tailored for graph-structured data. However, this is not convincing since the proposed measure is based on the learned embeddings (i.e., there is no graph structures). I think the general problem is still the same with the previous work. Hence, the novelty of SAD-Metric seems to be limited.

Minor:
-For model optimization, what does the $F$ stand for in Eq. (12)?
-The examples in the second paragraph of section 2.1 are redundant.

---

> ### Author Response · Authors · 2020-11-17
> **Official Response to AnonReviewer3**
>
> Thanks for your constructive reviews, which definitely help improve the quality of our work.
>
> The responses to your questions are as follows:
>
> Q1: Why a classifier-based SAD-Metric is designed? Can this metric lead to misjudgement? Why not designing a node-level measurement for structure-attribute disentanglement?
>
> A1: In our method, we design a classifier-based SAD-Metric, which is mainly due to two desiderata of such a metric: (1) it should measure the *dependency* between different channels of a graph embedding; (2) it should evaluate the *interpretability* of these channels, i.e. whether the channels uniquely correspond to the semantics of graph structure and node attribute. It is true that some classifier-free metrics (e.g. mutual-information-derived ones [a,b]) can effectively measure the feature dependency, while they can hardly judge the specific semantics related to each feature channel, i.e. fail to meet the second requirement.
>
> As a matter of fact, the proposed classifier-based metric can avoid the misjudgement when some structure (attribute) feature elements are correlated with the opposite part. In such a situation, the structure (attribute) element will be more sensitive to the change in node attribute (graph structure), which makes it harder to classify what kind of change happens on a graph, and thus leads to lower SAD-Metric score.
>
> Thanks for your good suggestion of designing a node-level metric! We have supplemented a node-centric structure-attribute disentanglement metric (node-SAD-Metric) in the **Section E** of appendix. This metric measures the extent of structure-attribute disentanglement in node embeddings. We experimentally find that the model's performance is comparable under the SAD-Metric and node-SAD-Metric. You can refer to the **Section E** of revised paper for more details.
>
> Q2: The novelty of the proposed SAD-Metric seems to be limited.
>
> A2: Compared with previous disentanglement metrics for images [a,b,c,d], the proposed SAD-Metric is novel on its input processing scheme. Specifically, we derive structure- or attribute-modified counterpart of each graph for measurement, which is tailored for graph-structured data. However, as you mentioned, the embedding-level evaluation of our method follows the previous works in the image domain. We believe improving such embedding-based measurement for graph-structured data will be an interesting direction of our future research.
>
> Q3: What does the $F$ stand for in Eq. (12)?
>
> A3: The $F$ in Eq. (12) stands for a neural network modelling the conditional distribution for edge reconstruction, which is stated in the *Learning structure embedding by edge reconstruction* part of **Section 3.2** and detailedly discussed in the **Section B** of appendix.
>
>
> **In the revised paper, you can refer to the bolded sections above for the detailed contents related to your questions.**
>
>
> [a] Chen, Ricky TQ, et al. "Isolating sources of disentanglement in variational autoencoders." NeurIPS, 2018.
>
> [b] Ridgeway, Karl, and Michael C. Mozer. "Learning deep disentangled embeddings with the f-statistic loss." NeurIPS, 2018.
>
> [c] Higgins, Irina, et al. "beta-vae: Learning basic visual concepts with a constrained variational framework." ICLR, 2017.
>
> [d] Kim, Hyunjik, and Andriy Mnih. "Disentangling by factorising." ICML, 2018.

---

> > ### Comment · AnonReviewer3 · 2020-11-23
> > **Still not convinced for Q1**
> >
> > Thanks for the detailed explanations from the authors. However, I am not convinced by the authors regarding my Q1. First, a classifier is not able to automatically learn which elements of the final graph embedding correspond to the structure/attribute factor, unless you explicitly design the classifier to let it know. So I think my proposed "misunderstanding example" still holds. Second, a carefully designed measure can naturally distinguish elements for the two semantic factors (structure or attribute).
> >
> > For Q2, the authors should revise the paper to precisely state the novelty of this work (the processing part, not the metric itself).

---

> > > ### Author Response · Authors · 2020-11-23
> > > **Official Response to AnonReviewer3**
> > >
> > > Thanks for your feedback!
> > >
> > > We respond to the two questions as follows:
> > >
> > > Q1: The classifier-based metric cannot judge whether an element of graph embedding corresponds to structure or attribute factor.
> > >
> > > A1: As you suggested, a classifier can hardly judge whether a feature element is for graph structure or node attributes. Therefore, we attempt a classifier-free variant of SAD-Metric, named as **SAD-Metric v2**, which is able to distinguish two types of factors for each feature element.
> > >
> > > Because of the limitation of time in the discussion period, we can only finish the evaluation on three methods (GCN-baseline, DisenGNN and Embed-SAD) using this metric, and the extensive evaluation on more methods will be done as soon as possible. Currently, the content about this new metric is not included in the paper due to its incompleteness. The detailed evaluation procedure and experimental results are as follows.
> > >
> > > **Evaluation procedure.** For evaluating SAD-Metric v2, we first modify the graph structure (node attributes) of each graph in the dataset and compute the absolute embedding difference of the graphs before and after modification (same as the $\Delta z_{\mathcal{G}}$ defined in **Section 3.3**), where the structure/attribute modification scheme follows that in SAD-Metric (**Section 3.3**). For each element in graph embedding, we identify whether it corresponds to structure or attribute factor based on the mean embedding difference of that element (i.e. the average of embedding difference over all graph samples). In specific, if the mean embedding difference induced by structure (attribute) modification is larger for an embedding element, we regard the element corresponds to graph structure (node attributes). After that, we compute the mean embedding difference of each element when modifying its irrelevant factor, and the average of such difference over all embedding elements serves as the score for SAD-Metric v2. For a model achieving good structure-attribute disentanglement, its score in terms of SAD-Metric v2 ought to be small.
> > >
> > >  **Results.** In Table A, we report the structure-attribute disentanglement score of three methods using SAD-Metric v2. It can be observed that the embedding elements extracted by the Embed-SAD model is least sensitive to the irrelevant factor, which demonstrates its superiority in disentangling the structure and attribute information in graph representations.
> > >
> > > Table A: The score of *SAD-Metric v2* on molecular graph datasets.
> > >
> > > |Method|BBBP|Tox21|ToxCast|SIDER|ClinTox|MUV|HIV|BACE|Avg|
> > > |:----:|:----:|:----:|:----:|:----:|:----:|:----:|:----:|:----:|:----:|
> > > |GCN-baseline|0.081|0.160|0.180|0.278|0.152|0.086|0.157|0.218|0.164|
> > > |DisenGNN|0.029|0.087|0.150|0.063|0.091|0.034|0.092|0.067|0.077|
> > > |Embed-SAD|**0.018**|**0.038**|**0.027**|**0.040**|**0.031**|**0.019**|**0.036**|**0.025**|**0.029**|
> > >
> > > Q2: The contribution of SAD-Metric should be precisely stated in the paper.
> > >
> > > A2: We have revised the paper, in which the exact contribution of SAD-Metric (i.e. the graph-specific data processing scheme) is stated in the contribution list of introduction (**Section 1**) and also in the related work part (**Section 4**).

---

### Official Review · AnonReviewer1 · 2020-10-28
**An interesting problem setup and method with clear intention, but with small questions for the experiments.**

**Rating:** 8
**Confidence:** 5

**Review:**

Summary: This paper presents a novel method called Embed-SAD (as well as Input-SAD) to learn graph/node representations to disentangle structure and attribute information. Input-SAD is a simple baseline that tries to get structure-attribute disentanglements by individually processing graph structures and node attributes. For structure, the original node attibutes are replaced by out-degrees only, and passed to GNNs, while for attibutes, the node attibutes are passed to fully-connected networks. Embed-SAD is a more elaborate method to disentangle the GNN embeddings by posing two types of additional losses, i.e., the edge-reconstruction loss for structures, and the Noise-Contrastive Estimation (NCE) loss to maximize the mutual information against the structure-encoding vectors, in addition to the original loss for supervision. The paper also develops an interesting evaluation metric called SAD-Metric where node attibutes or graph structures are exclusively perturbed for each graph, and prediction for whether that perturbation is for structure or for attibutes made by the element-wise absolute differences between embedded vectors before and after the perturbation. This SAD-Metric can quantify the extent to which the obtained representation can detect which perturbation, that for structures or that for attibutes, is made for each sample graph. The experimental results also demonstrated that the structure-attibute disentanglement by Embed-SAD learning strategy actually improved the prediction performance of many off-the-shelf GNNs over many different graph- or node-level tasks.

Comments:
The paper is well-written and easy to follow since the intention and corresponding ideas are quite clear. The combined loss fo Embed-SAD also has ablation study results that was also informative. Basically I liked the idea and have not too much to say, but here are some  comments:

- The SAD-metric would be more carefully evaluated since it includes "training" and we might be able to get zero-training error if embedding vectors are sufficiently high dimensional or we can use a strong predictor here for memorization. Even for a simple linear classifier (a "low capacity" classifier), I think it's better to replace this part by cross validation, or at least, a simple training-test split.

- This study would depend on what node attibutes (i.e. "featurization") we use, and it should be more clearly described in the main body what node attibutes are used in each benchmarking experiment. In particular, we often include "structure-derived features" such as degrees (or the number of attached hydrogens in the case of molecular data) to the node attributes of GNNs, and how this practice affects the disentanglement would be informative in the paper's context. In particular, for Input-SAD.

- Also, we need more information on the baselines of Table 3 for evaluating the SAD-metric. What is "Random-GCN (2017)"? (GCN with random node attibutes?) What is the intention to include this one here? Even Random-GCN had around 90% accuracy, and we need an additional baseline that clearly fails to disentangle structure and attibute information.

- First, I personally felt that Input-SAD is actually not for "disentanglement" and a bit confusing, rather it's a simple baseline to see what if we input structure- and attibute- information separetely in the first place. However, to my surprise, the performance of Input-SAD was not that bad for molecular graph classification benchmark, in a sharp contrast to those for citation and coauthor network cases. Actually, Input-SAD was better than Embed-SAD for SIDER and BACE. Are there any possible explanations on this?

- For eq (2), t_uv seemed undefined (though we can guess it). Do GIN, GAT, GraphSage use the edge attibutes?

---

> ### Author Response · Authors · 2020-11-17
> **Official Response to AnonReviewer1**
>
> Thanks for your appreciation of our work!
>
> The responses to your questions are as follows:
>
> Q1: A test split is required when evaluating the SAD-Metric.
>
> A1: For the evaluation of SAD-Metric on MoleculeNet (**Section 5.4**), we employ the scaffold split scheme to divide the whole dataset into training and test splits. The linear classifier is trained with the graphs in the training split and evaluated by the graphs in the test split, which avoids the possibility that the classification problem is solved by memorization.
>
> Q2: The node attributes used in each benchmark task should be stated in the paper.
>
> A2: Thanks for your suggestion! In the part of **Section 5.2 and 5.3** where we introduced the data sets, we have added the description of node attributes used in each benchmark dataset.
>
> Q3: More information is needed for the baseline model in the SAD-Metric experiment. Also, a weaker baseline should be added.
>
> A3: We apologize for the naming mistake in our original paper version. The 'random-GCN' in the original paper refers to the GCN model pre-trained on the graph classification task of corresponding molecule dataset, so we rename it as 'GCN-baseline' in the revised paper. To resolve your concern, we add a GCN model with random parameters, and it is named as 'random-GCN' in the revised paper due to its randomness of model parameters. This baseline model achieves only 75.6% average SAD-Metric score, which indicates that it fails to disentangle the structure and attribute information. You can refer to the **Section 5.4** and **Table 3** of revised paper for detailed model settings and experimental results.
>
> Q4: Why does the Input-SAD model perform much better in graph classification tasks than in node classification tasks?
>
> A4: For node classification, since the Input-SAD model directly disentangles a graph into a structure and an attribute component, some structure (attribute) information contained in the opposite component may be lost in node embeddings. However, for graph classification, the graph pooling operation is able to supplement the missing structure or attribute information of each node from other nodes, and thus leads to the decent performance of Input-SAD on molecular graph classification benchmarks.
>
> Q5: What does $t_{uv}$ in Eq. (1) mean? Do different GNNs use edge attributes in the proposed method?
>
> A5: In Eq. (1), $t_{uv}$ denotes edge attributes. In our experiments, we add the information of edge attributes to the information propagation process of GNN (including GCN, GraphSAGE, GAT and GIN).
>
>
> **In the revised version, you can refer to the bolded sections above for the detailed contents related to your questions.**

---

### Official Review · AnonReviewer2 · 2020-10-29
**The intuition, model and experimental results are good, but lacking solid theoretical support and analysis.**

**Rating:** 4
**Confidence:** 4

**Review:**

This paper focuses on disentangling embeddings of the structure and the attribute of graph. The authors' key idea is that the structure and attribute information should be split in GNN. Based on this, the authors try to disentangle the structure embedding and the attribute embedding. With two different components, two different kinds of embeddings can be captured at the input stage.   In addition, these two different kinds of embeddings can be obtained by reconstructing the edge and minimizing the mutual information. At last, the authors propose a metric to evaluate the disentanglement. The models in this paper outperform baselines in node classification and graph classification task.

The pros of this paper is as following:
1, The motivation/intuition of this paper is good. Current GNNs tend to aggregate the attribute information by graph structures, but ignore the relation between the structure and attribute information.
2, This paper brings out two models to disentangle the structure and attribute information, at the input stage and the embedding stage-INPUT-SAD and EMBED-SAD, which seem reasonable to some extent.
3, The experimental results are good on both node classification task and graph classification task.

The cons of this paper:
1, Key weakness: the authors raise the problem that aggregation way of structure and attribute in current GNNs but not analyze the problem theoretically. For example, the true relation of structure and attributes and how the model in this paper can be derived naturally. The makes the novelty of the paper incremental.
2, The authors use degree serves as attribute in INPUT-SAD. As I know, degree is a usually used attribute in graph. This design is contradictory to the authors' claim.

Question for authors: If some theoretical analysis about why the disentanglement is essential can be added into this paper. I may consider changing my reviews.

---

> ### Author Response · Authors · 2020-11-17
> **Official Response to AnonReviewer2**
>
> Thanks for your insightful comments and suggestions, which definitely help improve our work! We have revised the paper according to your suggestions.
>
> We respond to your concerns as follows:
>
> Q1: The theoretical analysis about the validity of structure-attribute disentanglement should be provided.
>
> A1: In **Section 3.4**, we add the theoretical analysis about the benefit of learning structure-attribute disentangled graph representations. In summary, we illustrate that such a disentanglement can reduce the search space of effective graph embeddings and hence allows a more effective optimization of graph encoders. Please refer to **Section 3.4** in the revised version for a detailed theoretical analysis.
>
> Q2: Does the use of degree feature in the structure component of Input-SAD violate the principle of the model?
>
> A2: In the Input-SAD model, in order to perform GNN-based information propagation on the structure component, we have to assign each node a feature. In our method, we use the out-degree of each node as its structure-derived feature/attribute, which does not violate the principle that the structure component contains only the information of graph structure.
>
>
> **In the revised paper, you can refer to the bolded sections above for the detailed contents related to your concerns.**

---

### Decision · Program_Chairs · 2021-01-07
**Final Decision**

**Decision:**

Reject

**Comment:**

In this paper, the authors propose a method to find disentangling embeddings of the structure and the attribute of the graph. Overall, this is an interesting paper and the paper is well-written and easy to follow, and the paper has some merits. However, the reviewers were still not convinced by the response, and the paper is still below the acceptance threshold.  I encourage authors to revise the paper based on the reviewer's comments and resubmit it to a future venue.